Effects of cadmium stress on seed germination and physiological-biochemical characteristics in okra: a comparative study of red and green varieties

Wang Wanwan
Yu Lei
Zhang Xuexia
Yu Jingbo
Jiang Zhou
Xu Long
Rui Haiyun ruihaiyun@tzu.edu.cn
Taizhou University, Jiangsu Key Laboratory of Chiral Pharmaceuticals Biosynthesis , Taizhou , Jiangsu , China
Abd El-Moneim Diaa
Electronic publication date: 2025 May 28
Publication date: 2025
Volume: 13
Electronic Location ID: e19498
Received 2025 Jan 10; Accepted 2025 Apr 29
Copyright: ©2025 Wang et al.
Copyright year: 2025
Copyright holder: Wang et al.
License: This is an open access article distributed under the terms of the Creative Commons Attribution License, which permits unrestricted use, distribution, reproduction and adaptation in any medium and for any purpose provided that it is properly attributed. For attribution, the original author(s), title, publication source (PeerJ) and either DOI or URL of the article must be cited.
License URL: https://creativecommons.org/licenses/by/4.0/

Keywords: Cd stress, Non-enzymatic antioxidant, Antioxidant enzyme, Red and green okra

Funding: The Taizhou Science and Technology Support Program (Agriculture) Project TN202222 Natural Science Foundation for Colleges and Universities in Jiangsu Province 22KJB360017 The Qing Lan Project of Jiangsu Province This study was funded by the Taizhou Science and Technology Support Program (Agriculture) Project (TN202222), Natural Science Foundation for Colleges and Universities in Jiangsu Province (22KJB360017), and the Qing Lan Project of Jiangsu Province. The funders had no role in study design, data collection and analysis, decision to publish, or preparation of the manuscript.

==============================
Previous study has demonstrated that okra (Abelmoschus esculentus (L.) Moench) is capable of accumulating cadmium (Cd) in both plants and fruits. However, there is still little known about the physiological responses of different okra cultivars to Cd accumulation. This study investigated the effects of exogenous Cd application on various growth parameters and physiological aspects in two okra varieties (red and green okra). The results indicated that Cd exposure had a dose-dependent inhibitory effect on seed germination rate and potential. Addition of 50 µM Cd to the hydroponic solution led to a significant reduction in plant biomass. In the red okra variety, Cd accumulation peaked at 287.8 mg kg−1 DW in shoots and 790.3 mg kg−1 DW in roots, while in green okra, these values reached 280.8 mg kg−1 DW in shoots and 903.7 mg kg−1 DW in roots. Furthermore, the Cd treatment resulted in an increase in chloroplastic pigment content of both okra varieties. Production of superoxide anion (O2•−) and hydrogen peroxide (H2O2) significantly rose in the roots of both varieties, with O2•− levels increasing by 298.8% in red okra and 505.8% in green okra roots, and H2O2 levels increasing by 2.23-fold in red okra and 1.4-fold in green okra roots. This rise in reactive oxygen species led to elevated proline content (3.09-fold in red okra roots and 8.45-fold in green okra roots) and non-protein sulfhydryl (NPSH) levels (13.52-fold in red okra roots and 10.21-fold in green okra roots), as well as increased activities of peroxidase (POD) and superoxide dismutase (SOD). Notably, green okra exhibited a more efficient antioxidant defense system and greater tolerance to Cd-induced oxidative stress compared to red okra. This study provides a foundation for developing Cd-tolerant crop varieties and improving phytoremediation strategies.

Introduction

The growing problem of heavy metal pollution in soil is a direct consequence of the irrational exploitation of mineral resources, coupled with the detrimental effects of factors such as sewage irrigation, the improper use of fertilizers and livestock farming (Bhuiyan et al., 2010; Liu et al., 2016). Among these heavy metals, cadmium (Cd) is regarded as one of the most toxic. Long-term irrigation with untreated wastewater can increase soil Cd content by 2–5 times, reaching levels as high as 1.85 mg/kg (Zhang et al., 2019). Long-term use of phosphate fertilizers has resulted in Cd accumulation rates of 0.05–0.15 mg/kg per year (Loganathan, Hedley & Grace, 2008). The increasing release of Cd into the environment is a growing concern as it presents an ever-increasing risk to human health (Wang et al., 2015). The Joint FAO/WHO Expert Committee on Food Additives (JECFA) has set a provisional tolerable weekly intake (PTWI) for Cd at 2.5 µg/kg body weight per week. Cd is non-essential for plant growth and development, and its toxicological threshold levels vary significantly across different plant species. It was found that Cd concentrations above 5 mg/kg significantly reduced growth and photosynthesis in rice plants (Das, Samantaray & Rout, 1997). When exposed to excess Cd, plants may exhibit symptoms such as reduced seed germination, inhibited root elongation and reduced photosynthetic activity (Bae, Benoit & Watson, 2016; Soudek et al., 2014). Moreover, Cd accumulation in plant tissues can disturb nutrient uptake and translocation, disrupt enzymatic activities, and induce the generation of reactive oxygen species (ROS), including superoxide anion (O2•−), hydrogen peroxide (H2O2), and hydroxyl radicals (•OH) (Rizwan et al., 2017; El Rasafi et al., 2022; Song, Jin & Wang, 2017). The generated ROS can disrupt calcium signaling, attack polyunsaturated fatty acids in cell membranes, oxidize proteins, and cause DNA strand breaks and base modifications (Rizwan et al., 2017). These effects ultimately impair plant growth and can compromise crop yield and quality. Given the widespread occurrence of Cd contamination in agricultural soils, elucidating the specific mechanisms underlying Cd toxicity in plants is crucial for devising effective management strategies to mitigate its harmful effects.

Plants have evolved sophisticated morphological and physiological defense mechanisms to cope with Cd stress and minimize its toxic effects. These strategies can be categorized into two main approaches: (1) restricting metal uptake to prevent accumulation to toxic levels, and (2) reducing metal toxicity by accumulating, storing, and immobilizing the harmful elements in specific plant tissues (El Rasafi et al., 2022). When Cd enters plant cells, it activates the plant’s immune response. A crucial response entails the activation of the antioxidant defense system, which encompasses non-enzymatic antioxidants such as glutathione and phytochelatins, alongside various antioxidant enzymes including peroxidase (POD) and superoxide dismutase (SOD) (Huang et al., 2019; Meng et al., 2019; Rui et al., 2016). This intricate antioxidant system is activated to preserve the redox equilibrium of plant cells by transforming ROS into less harmful products. In addition, plants may enhance the expression of stress-responsive genes involved in metal transport, detoxification, and cellular repair pathways (Liu et al., 2017; Qiao et al., 2019; Rui et al., 2018).

In the context of scarce arable land resources worldwide, effective remediation of heavy metal contamination in soil is an urgent issue. Phytoremediation technology, as a novel soil remediation approach, offers numerous advantages such as low cost, wide applicability, ease of operation and the potential to reduce secondary contamination, making it one of the most important soil remediation techniques at present (Liu et al., 2018; Vassilev et al., 2004). However, its effectiveness is constrained by factors such as contaminant depth, plant specificity, and time requirements. For optimal results, phytoremediation is often used in conjunction with other remediation technologies. There is currently a lot of research into hyperaccumulating plants, which are plants capable of absorbing one or more heavy metals at levels hundreds or thousands of times higher than typical plants (Rascio & Navari-Izzo, 2011; Vassilev et al., 2004). Over 700 species of hyperaccumulating plants have been identified globally, spanning across 50 families (Reeves et al., 2018). Presently, there are over seven Cd hyperaccumulating plants that can be utilized, most of which are Brassicaceae and Crassulaceae (Stein et al., 2017; Qiu et al., 2012). Most of the Cd hyperaccumulating plants discovered so far have limited biomass and are restricted in their growth season, posing practical limitations. Therefore, it is of practical significance to research and explore heavy metal-accumulating plants with larger biomass, stronger adaptability, and wider ecological ranges. Okra (Abelmoschus esculentus (L.) Moench) is a versatile and high-value herbaceous plant (Dantas, Alonso & Florentino, 2021). Red okra, a variant with red or purple pods, yields approximately 3,000 to 5,000 kg per hectare and is primarily cultivated in South Asia and West Africa (Benchasri, 2012). It is particularly popular among ethnic groups in South India and West Africa, where it is used in traditional dishes such as sambar, curries, and soups. There is a risk of Cd contamination in okra growing areas due to improper use of phosphate fertilizers and sewage irrigation. In Cd-contaminated soils, the stems of okra contain up to 5–10 mg/kg dry weight (DW) of Cd, while the roots usually contain higher levels of Cd, up to 7–14 mg/kg DW (Sharma, Agrawal & Agrawal, 2010a; Sharma, Agrawal & Agrawal, 2010b). In contrast, the hyperaccumulator Solanum nigrum exhibits significantly greater Cd accumulation in its above-ground tissues, reaching concentrations of 124.6 mg/kg DW (Sun et al., 2007), which substantially exceeds the accumulation capacity observed in okra. However, the advantages of okra lie in its economic value and food safety, making it potentially useful for remediation of mildly Cd-contaminated soils and for agricultural production.

Although studies have reported Cd accumulation mechanisms in okra, no research has compared responses across varieties. In the present study, we investigated the responses of two okra varieties with different fruit colors (red and green) to external Cd treatment. The study focused on seed germination, seed growth, plant biomass, ROS production, antioxidant enzyme activities, non-enzymatic antioxidants content, and chloroplastidic pigment content. This was done to better understand Cd tolerance in these varieties. Our findings could provide valuable insights for phytoremediation strategies and advance the scientific understanding of plant-metal interactions, which could inform strategies to mitigate the effects of environmental pollution on agricultural ecosystems and human health.

Materials and Methods

Determination of germination rate and germination potential

Seeds of red and green okra (purchased from Hebei Nanjixing Seed Company) were sterilized in 3% H2O2 for 15 min, rinsed thrice with distilled water, and blotted dry. Cd solutions (0, 0.5, 1.0, 1.5, 2.0 mM) were prepared by diluting 0.1 M CdCl2•2.5H2O stock. For 2.0 mM solution, 40 mL of stock was diluted to 2 L with distilled water. Each 9-cm-diameter petri dish (lined with two layers of filter paper) was saturated with a Cd solution. Thirty seeds were placed per dish (n = 3 replicates per treatment) and incubated at 24 °C in darkness. The number of germinated seeds was recorded each day for 7 days (germination was defined as radicle emergence ≥ 2 mm) and calculated the seed germination potential and germination rate. The formula for the germination potential (GP) is given by the following equation, GP%=G3Gn×100

where GP is the 3-day germination count (G3) divided by the tested count (Gn). And the formula for the germination rate (GR) is given by the following equation, GR%=G7Gn×100

where GR is the 7-day germination count (G7) divided by Gn.

Plant growth and treatment

The seeds germinated in a plastic container (length 28 cm, width 20 cm, height 10 cm) with vermiculite. Water was added once daily to keep the vermiculite moist. The 7-day-old seedlings were transferred to 1-litre flasks filled with Hoagland’s nutrient solution, with 10 plants in each. After 3 days, the Hoagland nutrient solution in pots was replaced by a new nutrient solution containing 0 or 50 µM Cd (added as CdCl2⋅2.5H2O). The choice of 50 µM Cd was based on previous research on Cd stress in Vicia sativa (Rui et al., 2016), where this concentration was found to be effective in inducing measurable physiological and biochemical responses without causing extreme toxicity that could mask specific stress-related effects. Plant growth and treatment experiments were all conducted in a greenhouse at 26 ± 2 °C, 70–80% relative humidity and 14 h photoperiod.

Biomass determination

After 6 days of Cd exposure, plant phenotypes were observed and photographed, and lengths of longest roots and shoots were recorded. The roots and shoots of the seedlings were collected separately. The tissues were thoroughly rinsed with deionized water three times and blotted dry with absorbent paper. For moisture content determination, fresh weight (FW) was immediately measured using an analytical balance after surface moisture removal. Samples were then oven-dried at 80 °C for 24 h until constant weight was achieved. Dry weight (DW) was recorded after cooling samples in a desiccator containing silica gel for 30 min. Moisture content (%) was calculated as: [(FW–DW)/FW] ×100. Each plastic cup contained ten seedlings as a replicate. Three plastic cups, or 30 seedlings, were randomly selected for one treatment.

Determination of Cd content

Roots and shoots were harvested separately and baked at 80 °C for 24 h. The dried samples were ground into powder and passed through a 60-mesh sieve (with a pore size of approximately 0.25 mm). Five milliliters of nitric acid and 0.1 mL of internal standard mix were added to 200 mg of powder. After 1 h, the mixture was diluted to 25 mL and then digested according to the standard procedure of the microwave digestion apparatus. The digestion process was carried out by gradually raising the temperature to 120 °C and maintaining it for 5 min, followed by further increasing the temperature to 160 °C and holding it for 10 min, and finally elevating the temperature to 180 °C and maintaining it for 20 min. After completion of the digestion process, the samples cooled to room temperature. The samples were determined by inductively coupled plasma mass spectrometry (ICP-MS). For quality assurance, a blank solution and certified Cd were analyzed after every ten samples to verify the accuracy and precision of the measurements. The internal standard (10 µg/L) was continuously introduced through a separate channel to correct for potential signal drift and matrix effects, see Rui et al. (2016) for details. The samples were quantified by calculating the ratio of the intensity of the mass spectral signals of Cd to those of the internal standard, following the established calibration curve. The Cd translocation factor (TF) was calculated based on the Cd content in plant roots and stems: TF%=CsCr×100

where TF is the percentage of shoot (Cs) to root (Cr) Cd concentration. This factor provides insights into a plant’s ability to absorb heavy metals from the soil and transport them to above-ground tissues.

Chloroplastidic pigment analysis

The roots and leaves of 10 seedlings in one pot were harvested separately 6 days after Cd treatment as a replicate for analysis of chloroplastidic pigments and other substances below. The extraction, detection and content analysis of chloroplastidic pigments are described in Zayneb et al. (2015). Briefly, 200 mg of fresh leaves were rapidly ground with two mL of pre-cooled 95% ethanol in combination with 0.1% CaCO3 to prevent chlorophyllase activity. The extract was filtered and diluted to 25 mL with 95% ethanol. Absorbance values were determined at 665 nm, 646 nm and 470 nm using 95% ethanol as a blank.

H2O2 and O2•− analysis

For H2O2 measurement, one mL of pre-cooled acetone was added to 100 mg of liquid nitrogen-ground samples. Then the mixture was centrifuged at 12,000 rpm at 4 °C for 20 min, and the supernatant was collected. The H2O2 levels were assessed in accordance with the guidelines provided by the Hydrogen Peroxide Content Assay Kit (Sangon Biotech, #D799774, Shanghai, China). For O2•− analysis, one mL of pre-chilled phosphate-buffered saline (pH 7.8) was added to 100 mg of cryogenically ground samples. Following centrifugation under the same conditions, the supernatant was collected. The O2•− producing rate in root and leave extracts were evaluated following the protocol outlined in the Superoxide Anion Content Assay Kit (Sangon Biotech, #D799771, Shanghai, China).

SOD and POD analysis

The frozen samples (100 mg) ground in liquid nitrogen were thawed on ice, and one mL of 0.05 M potassium phosphate buffer (PPB, pH 7.8) was added. After homogenization, the mixture was centrifuged at 12,000 rpm at 4 °C for 20 min, and 200 µL of supernatant was collected for SOD activity measurement. The reaction system for the POD activity of one sample determination contained 2.7 mL of 14.5 mM methionine, 0.1 mL of 3mM EDTA-Na2, 0.1 mL of 60 µM riboflavin, 0.1 mL of 2.25 mM nitro-blue tetrazolium (NBT) and 40 µL of supernatant. After mixing, the tubes were incubated under 4,000 lux at 25 °C for 20 min. Absorbance at 560 nm (OD560) was measured using a SpectraMax M5 microplate spectrophotometer (Molecular Devices, San Jose, CA, USA), with an unilluminated tube as blank control. Commercial SOD (Sigma-Aldrich, St. Louis, MO, USA 100 U/mL) served as positive control, while a sample-free reaction was included as negative control.

The frozen samples (100 mg) ground in liquid nitrogen were thawed on ice, and one mL of 0.1 M potassium phosphate buffer (PPB, pH 7.0) containing 1% polyvinylpyrrolidone (PVP) was added. After homogenization, the mixture was centrifuged at 12,000 rpm at 4 °C for 20 min, and the supernatant was collected for POD activity measurement. For POD activity determination, the reaction mixture consisted of 1.8 mL of 0.1 M HAC buffer (pH 5.4), one mL of 0.75% guaiacol, 50 µL of supernatant, and 0.1 mL of 3% H2O2. After mixing, absorbance at 460 nm was recorded every 15 s for 3 min.

NPSH analysis

The extraction and content analysis of non-protein sulfhydryl compounds (NPSH) were performed according to Morelli & Scarano (2004) with a slight modification. Briefly, roots or leaves (about 500 mg each) were ground in liquid nitrogen and one mL of extraction solution containing 5% (w/v) sulfosalicylic acid and 6.3 mM diethylenetriaminepentaacetic acid (DTPA) was added to each sample. The mixture was centrifuged (12,000 rpm) at 4 °C for 15 min, and the supernatants were collected. The reaction system for NPSH analysis contained 2.1 mL of 0.5 M K2HPO4, 85 µL of 10 mM 5,5′-Dithiobis-(2-nitrobenzoic acid) (DTNB) and one mL of supernatant. After reacting at 30 °C for 2 min, absorbance at 412 nm was measured at 412 nm. Results were plotted against a standard curve generated from one mM reduced GSH.

Proline analysis

Roots or leaves (approximately 100 mg each) were homogenized in liquid nitrogen, followed by the addition of one mL of a 3% (w/v) sulfosalicylic acid solution to each sample. The samples were then subjected to boiling water for 10 min before centrifugation at 8,000 rpm for 15 min, after which the supernatant was carefully collected. To this supernatant, 400 µL of glacial acetic acid and 400 µL of acidic ninhydrin were added and reacted in boiling water for an additional 30 min. After cooling, 600 µL of toluene was added and the mixture was vortexed for 30 s. The upper phase was collected and centrifuged at 3,000 rpm for 5 min (Nayyar & Walia, 2003). Absorbance was subsequently measured at a wavelength of 520 nm using toluene as a blank. Proline content was determined using a standard curve (0–100 µM) prepared from 1 mM L-proline standard (Sigma-Aldrich, St. Louis, MO, USA ≥99% purity), with each concentration assayed in triplicate.

Data analysis

The data were analyzed using Statistica 6. Results of all the bar graphs are the average of three independent determinations + SE. Differences in the levels of Cd and endogenous substances were analyzed by Student’s t test. Differences in germination rate and germination potential among different concentrations of Cd were analyzed by one-way ANOVAs (P < 0.05; Duncan’s multiple range test).

Results

Cd treatment inhibited seed germination

To explore if Cd exposure affect the seed germination of okra, we conducted the experiments on GR and GP with Cd at different concentrations ranging from 0.5 to 2 mM. Exogenous application of Cd inhibited GR and GP of okra in a dose-dependent manner (Figs. 1C, 1D). For red okra, all concentrations significantly reduced GR by the 7th day (Figs. 1A, 1B). Similarly, in green okra, all concentrations except 0.5 mM significantly inhibited seed germination (Figs. 1A, 1B). Notably, at two mM Cd exposure, the germination rates were markedly reduced to 53% (95% CI and p < 0.001) for red okra and 48% (95% CI and p < 0.001) for green okra.

Figure 1 Effect of Cd treatment on seed germination of okra.

(A and B) Germination rate (A) and germination potential (B) of red okra and green okra treated with 0 to 2 mM Cd. (C and D) Seedling morphology of red okra (C) and green okra (D) after treated with 0 to 2 mM Cd for 7 d. Lower and upper case letters indicate significant differences of red okra and green okra among treatments, respectively (p < 0.05; Duncan’s multiple range test).

The Cd accumulation in okra plant

After subjecting both red and green okra seedlings to a 6-day treatment with 50 µM Cd, significant Cd accumulation was observed in the shoots and roots of both okra varieties. In red okra, the Cd content peaked at 287.8 mg kg−1 DW in the shoots and 790.3 mg kg−1 DW in the roots (Fig. 2A). In green okra, the Cd content reached 280.8 mg kg−1 DW in the shoots and 903.7 mg kg−1 DW in the roots (Fig. 2B). Calculations revealed that the translocation factor (TF) of Cd stood at 49.24% in red okra and 30.73% in green okra. The TF metric signifies the percentage of Cd effectively translocated from the roots to the shoots within the plant system.

Figure 2 The Cd content in okra plants.

The Cd content in red okra (A) and green okra (B) grown in nutrient solution containing 50 µM Cd for 6 d. (C) TF of green okra and red okra. TF (translocation factor) is the ratio of Cd concentration in leaves to Cd concentration in roots.

Cd treatment inhibited plant growth

To explore the effect of Cd on plant growth, we conducted the experiment adding 50 µM Cd to the nutrient solution. As shown in Fig. 2, Cd treatment significantly inhibited the growth of both red okra (Fig. 3A) and green okra (Fig. 3B). The length of shoots of red and green okra decreased by 20.3% (95% CI and p = 0.02) and 25.9% (95% CI and p = 0.0013) (Fig. 4A, Left), respectively, while the length of their roots decreased by 45.3% (95% CI and p < 0.001) and 27.0% (95% CI and p < 0.001) (Fig. 4A, Right). We also tested the dry and fresh weight of okra. Compared with the controls, both varieties exhibited a significant reduction in fresh weight (Fig. 4B) and dry weight (Fig. 4C). The moisture content of okra plants is almost unaffected by Cd treatment, with only the roots of Cd-treated green okra being 1.9% (95% CI and p = 0.045) lower than the control (Fig. 4D).

Figure 3 Effect of Cd treatment on growth of okra plants.

Growth phenotypes of leaves and roots of red okra (A) and green okra (B) grown in nutrient solution containing 50 µM Cd for 6 d.

Figure 4 Effect of Cd treatment on biomass of okra plants.

The length (A), fresh weight (B), dry weight (C) and moisture content (D) of shoot (Left) and root (Right) of red okra and green okra grown in nutrient solution containing 50 µM Cd for 6 d. Asterisks indicate significant differences between Cd and control (**P < 0.05, ***P < 0.05, Duncan’s multiple range test).

Cd treatment increased chloroplastidic pigment content

The impact of Cd treatment on chloroplastidic pigment content in both red okra and green okra was explored. The results demonstrated that Cd treatment elicited distinct responses in the levels of chloroplastidic pigments in two plant varieties studied. Specifically, in red okra, Cd application resulted in a significant 14.4% increase (95% CI and p = 0.011) in chlorophyll a content and a 32.4% increase (95% CI and p = 0.05) in carotenoid content, whereas no significant effect was observed on chlorophyll b level (Fig. 5A). However, in green okra, Cd treatment led to a significant increase in the levels of all three pigments: chlorophyll a (37.4%, 95% CI and p = 0.002), chlorophyll b (46.4%, 95% CI and p < 0.001), and carotenoids (42.1%, 95% CI and p = 0.006) (Fig. 5B). Also, the increase in the chloroplastidic pigments was greater in green okra than in red okra.

Figure 5 Effect of Cd treatment on the chloroplastidic pigment content of okra plants.

The content of chloroplastidic pigments in leaves (A) and roots (B) of red okra and green okra grown in nutrient solution containing 50 µM Cd for 6 d. Asterisks indicate significant differences between Cd and control (**P < 0.05, ***P < 0.01, Duncan’s multiple range test).

Cd treatment increased H2O2 and O2•− production

We examined changes in H2O2 and O2•− levels under Cd stress. Treatment with 50 µM Cd significantly increased H2O2 levels in the roots of both red and green okra, with red okra exhibiting a 2.23-fold (95% CI and p = 0.03) increase and green okra a 1.4-fold (95% CI and p = 0.035) increase compared to controls (Fig. 6B), but not in their leaves (Fig. 6A). It is worth noting that red okra leaves contain higher levels of H2O2 than green okra. After 6 days of Cd treatment, O2•− production showed significantly increased level compared to control, O2•− production in leaves rose by 182.3% (95% CI and p < 0.001) in red okra and 120.9% (95% CI and p = 0.085) in green okra, while root activity surged by 298.8% (95% CI and p < 0.001) in red okra and 505.8% (95% CI and p < 0.001) in green okra (p < 0.05, 95%, Fig. 6D). Notably, red okra leaves exhibited higher O2•− production than green okra under the same Cd concentration (Fig. 6C).

Figure 6 Effect of Cd treatment on H2O2 content and O2•− producing rate of okra plants.

(A and B) The content of H2O2 in leaves (A) and roots (B) of red okra and green okra grown in nutrient solution containing 50 µM Cd for 6 d. (C and D) The producing rate of O2•− of leaves (C) and roots (D) of red okra and green okra grown in nutrient solution containing 50 µM Cd for 6 d. Asterisks indicate significant differences between Cd and control (**P < 0.05, ***P < 0.01, Duncan’s multiple range test).

Cd treatment increased the activity of antioxidant enzymes

In Cd-treated red and green okra, both POD and SOD showed significantly increased activity compared to controls. POD activity in leaves rose by 58.2% (95% CI and p = 0.001) in red okra and 233.7% (95% CI and p < 0.001) in green okra (Fig. 7A), while root activity surged by 959.3% (95% CI and p = 0.003) in red okra and 1,461% (95% CI and p < 0.001) in green okra (Fig. 7B). Similarly, SOD activity in roots increased by 229.3% (95% CI and p = 0.021) in red okra and 457.8% (95% CI and p < 0.001) in green okra (Fig. 7D). In contrast, leaf SOD activity showed only a modest 1.60-fold (95% CI and p = 0.029) increase in green okra, with minimal change in red okra (Fig. 7C). These results highlight a stronger Cd-induced antioxidant response in roots compared to leaves.

Figure 7 Effect of Cd treatment on POD and SOD activity of okra plants.

(A–B) The POD activity in the leaves (A) and roots (B) of red okra and green okra grown in nutrient solution containing 50 µM Cd for 6 d. (C–D) The SOD activity in the leaves (C) and roots (D) of red okra and green okra grown in nutrient solution containing 50 µM Cd for 6 d. Asterisks indicate significant differences between Cd and control (**P < 0.05, ***P < 0.01, Duncan’s multiple range test).

Cd treatment increased proline content

In red okra, proline content increased significantly only in the roots, reaching 3.09 (95% CI and p < 0.001) times higher than the control (Fig. 8B). In contrast, green okra showed elevated proline in both roots and leaves, with levels 8.45 (95% CI and p < 0.001) and 6.11 (95% CI and p = 0.003) times higher than controls, respectively (Figs. 8A, 8B). These results highlight varied proline modulation under Cd stress between the two varieties.

Figure 8 Effect of Cd treatment on proline content of okra plants.

The proline content in leaves (A) and roots (B) of red okra and green okra grown in nutrient solution containing 50 µM Cd for 6 d. Asterisks indicate significant differences between Cd and control (**P < 0.05, ***P < 0.01, Duncan’s multiple range test).

Cd treatment increased NPSH levels

After 6 days of Cd treatment, both red and green okra showed significant NPSH induction in the roots, with red okra exhibiting a 13.52-fold (95% CI and p = 0.006) increase and green okra a 10.21-fold (95% CI and p < 0.001) increase compared to controls (Fig. 9B). In contrast, leaf NPSH levels were less affected, with red okra showing a 1.25-fold (95% CI and p = 0.028)) increase and green okra a 1.82-fold (95% CI and p < 0.001) increase (Fig. 9A). These results indicate a stronger Cd-induced NPSH response in roots than in leaves.

Figure 9 Effect of Cd treatment on NPSH content of okra plants.

The content of NPSH in leaves (A) and roots (B) of red okra and green okra grown in nutrient solution containing 50 µM Cd for 6 d. Asterisks indicate significant differences between Cd and control (**P < 0.05, ***P < 0.01, Duncan’s multiple range test).

Discussion

Heavy metal contamination, a prevalent environmental problem, poses a significant threat to plant seed germination and seedling growth. Over the past four decades, extensive research has elucidated the adverse effects of heavy metal stress on plant seeds, leading to impaired germination rates and compromised seedling establishment. Our research findings indicate that cadmium (Cd) stress significantly reduces the seed germination rate in okra, which is consistent with previous studies on the effects of heavy metal stress on seed germination in various plant species. The inhibition of seed germination by Cd is often dose-dependent, as observed in pea (Pisum sativum) (Majeed, Muhammad & Siyar, 2019), fenugreek (Trigonella foenum-graecum) (Zayneb et al., 2015), rape (Brassica napus) (Rao et al., 2019), barley (Hordeum distichum) (Nouri, El Rasafi & Haddioui, 2019), wheat (Triticum aestivum) (El Rasafi et al., 2016), Peganum harmala (Nedjimi, 2020). These studies have shown that excessive concentrations of Cd significantly reduce germination rates, leading to poor seedling emergence and compromised plant growth. These studies, along with our results, suggest that Cd-induced inhibition of seed germination is a common response across various plant species. However, the extent of germination reduction may vary depending on the plant species, Cd concentration, and experimental conditions. Our study contributes to this body of knowledge by providing specific data on different varieties of okra. Previous studies have shown that Cd exposure reduces shoot and root length, fresh and dry weight, and overall leaf area (Nouri, El Rasafi & Haddioui, 2019; Rao et al., 2019; Majeed, Muhammad & Siyar, 2019). Consistent with these researches, this study observed similar decreases in okra under Cd stress. Cd stress significantly inhibited the growth of both varieties, with red okra exhibiting a greater reduction in shoot length (20.3%) compared to green okra (25.9%). In contrast, root length was more severely affected in red okra (45.3% reduction) than in green okra (27.0% reduction). This suggests that red okra roots are more sensitive to Cd toxicity, which may be linked to their higher Cd accumulation and subsequent oxidative damage. The stronger inhibition of root growth in red okra likely compromises water and nutrient uptake, indirectly affecting shoot growth. Interestingly, the moisture content of both varieties remained largely unchanged, indicating that Cd toxicity primarily affects biomass accumulation rather than water balance.

The reduction in biomass in Cd-treated plants may be attributed to inhibition of biosynthesis process. Cd has been discovered to have a significant impact on the process of photosynthesis in plants (He et al., 2017). This toxic metal interferes with enzyme activities and disrupts photosynthetic electron transport in a wide range of plant species. The presence of Cd can hinder the plants’ ability to efficiently convert light energy into chemical energy, ultimately affecting their overall growth and development (Bahmani, Modareszadeh & Bihamta, 2020). The main pigments involved in the absorption and transfer of light energy in photosynthesis in higher plants are chlorophyll and carotenoids. A lot of studies showed that exposure to Cd caused a decrease in chlorophyll content in various plant species, such as fenugreek (Zayneb et al., 2015), bean (Phaseolus vulgaris) (Bahmani, Modareszadeh & Bihamta, 2020), wheat (El Rasafi et al., 2016), and carrot (Daucus carota) (Sharma, Agrawal & Agrawal, 2010a; Sharma, Agrawal & Agrawal, 2010b). Sharma, Agrawal & Agrawal (2010a) and Sharma, Agrawal & Agrawal (2010b) found that Cd caused 38, 65, 43 and 32% reductions in chlorophyll a, b, total and carotenoid contents in okra plant grown in Cd contaminated soil (100 mg L−1). However, our results revealed that red and green okra plants grown in Cd-containing hydroponic solution exhibited increased chloroplastidic pigment levels. This divergence likely reflects complex interactions between developmental stage, growth environment and genotype-specific stress responses, mediated by dynamic biochemical and molecular adaptations. We attribute the observed differences to two main factors. Firstly, there is a disparity in the age of the okra plants used in our study compared to that of Sharma, Agrawal & Agrawal (2010a). In our study, okra plants were treated with cadmium at 3 days post-germination and exposed for 6 days. In comparison, Sharma, Agrawal & Agrawal (2010a) conducted their experiments on okra plants at 15 days post-germination, with a longer Cd exposure period of 30 days. It is possible that younger plants may prioritize rapid chlorophyll synthesis to establish photosynthesis, while older plants could face cumulative Cd damage to chloroplast ultrastructure. This hypothesis requires further investigation through additional experiments. Secondly, we acknowledge that differences in growth environment influence the response mechanisms of cadmium stress in okra. Hydroponics deliver bioavailable Cd2+ directly to roots, potentially stimulating the expression of genes involved in chlorophyll biosynthesis, such as HEMA1 (glutamyl-tRNA reductase) or CHLH (Mg-chelatase subunit H), whereas soil Cd complexes reduce uptake kinetics. The increase in chlorophyll content could represent a compensatory response to maintain photosynthetic efficiency under stress: (1) Elevated chlorophyll levels might enhance light absorption, compensating for potential damage to the photosynthetic apparatus; (2) Cd stress could potentially increase the energy demand for detoxification processes, prompting the plant to boost chlorophyll content to meet these needs. In addition, carotenoids, which play a role in photoprotection and antioxidant defense, may increase to mitigate Cd-induced oxidative stress. This could indirectly stabilize chlorophyll molecules and enhance their content. The differential responses between red and green okra (e.g., chlorophyll b increase only in green okra) suggest that genetic factors play a significant role in modulating chlorophyll metabolism under Cd stress. Green okra’s chlorophyll b increase may arise from higher CAO (chlorophyll a oxygenase) activity, which converts chlorophyll a to b. Red okra’s anthocyanin-rich phenotype could divert resources away from chlorophyll b synthesis, as anthocyanins compete for shared precursors (e.g., phenylalanine) in the shikimate pathway. Further research is needed to confirm these hypotheses.

In addition to morphological inhibition, Cd stress also causes physiological changes in plants, in particular ROS overproduction (Schützendübel et al., 2001; Gill & Tuteja, 2010; He et al., 2017). Cd stress induced significant oxidative stress in both varieties, as evidenced by increased H2O2 and O2•− production, particularly in the roots. Notably, red okra leaves exhibited higher O2•− production than green okra, indicating a stronger oxidative burst in the aerial parts of red okra. This may explain the greater growth inhibition observed in red okra shoots. Both varieties activated their antioxidant defense systems, with significant increases in POD and SOD activities. However, the response was more pronounced in green okra, particularly in the roots. These findings suggest that green okra has a more efficient antioxidant defense system, which may contribute to its better tolerance to Cd stress. In addition to elevated antioxidant enzyme activities, the levels of non-enzymatic antioxidants proline and NPSH were significantly increased in both varieties. Proline and NPSH compounds play crucial roles in osmotic adjustment and metal detoxification (Sun et al., 2007; Zayneb et al., 2015; Semida, Hemida & Rady, 2018; Rady et al., 2019). Under Cd stress, red okra showed a significant increase in proline content only in the roots (3.09-fold), whereas green okra exhibited elevated proline levels in both roots (8.45-fold) and leaves (6.11-fold). This indicates that green okra employs a more systemic osmotic adjustment strategy, which may help maintain cellular homeostasis and protect against Cd-induced damage. Similarly, NPSH levels increased more dramatically in the roots of both varieties, with red okra showing a 13.52-fold increase compared to 10.21-fold in green okra. This suggests that red okra relies more heavily on NPSH compounds for Cd detoxification in the roots, while green okra may utilize additional mechanisms, such as vacuolar sequestration or phytochelatin synthesis.

The differential responses of red and green okra to Cd stress highlight the importance of genetic variation in stress tolerance mechanisms. Green okra demonstrated superior antioxidant activity, systemic proline accumulation, and enhanced protection of photosynthetic pigments, which may collectively contribute to its better performance under Cd stress. In contrast, red okra exhibited higher Cd translocation, greater oxidative stress in leaves, and a more localized proline response, making it more susceptible to Cd toxicity. These findings suggest that green okra may be more suitable for cultivation in Cd-contaminated environments, whereas red okra could serve as a model for studying Cd translocation and its implications for food safety.

Conclusion

In conclusion, our results reveal that Cd stress adversely affects seed germination and plant growth in okra, with significant Cd accumulation in plant tissues. However, the two okra varieties exhibit distinct physiological and biochemical responses to Cd stress, with green okra demonstrating a more robust antioxidant defense system and greater tolerance to Cd-induced oxidative stress. By elucidating the unique physiological and biochemical responses of red and green okra to Cd stress, this study provides a foundation for developing Cd-tolerant crop varieties and improving phytoremediation strategies.

Supplemental Information

Supplemental Information 1 Raw data

We thank Jinghui Zhou and Keming Shan for helping to look after the okra plants.

Additional Information and Declarations

Competing Interests

Author Contributions

Data Availability

The authors declare there are no competing interests.

Wanwan Wang conceived and designed the experiments, analyzed the data, prepared figures and/or tables, authored or reviewed drafts of the article, and approved the final draft.

Lei Yu performed the experiments, prepared figures and/or tables, and approved the final draft.

Xuexia Zhang performed the experiments, analyzed the data, prepared figures and/or tables, and approved the final draft.

Jingbo Yu performed the experiments, prepared figures and/or tables, and approved the final draft.

Zhou Jiang performed the experiments, prepared figures and/or tables, and approved the final draft.

Long Xu analyzed the data, prepared figures and/or tables, and approved the final draft.

Haiyun Rui conceived and designed the experiments, authored or reviewed drafts of the article, and approved the final draft.

The following information was supplied regarding data availability:

The raw measurements are available in a Supplemental File.

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
