# Peer review of "Effects of cadmium stress on seed germination and physiological-biochemical characteristics in okra: a comparative study of red and green varieties"

_PeerJ, doi:10.7717/peerj.19498_

## Round 0.1 · original submission · Major Revisions

Dear Authors
The manuscript cannot be accepted for publication in its current form. It needs a major revision before publication. The authors are invited to revise the paper considering all the suggestions made by the reviewers. Please note that the requested changes are required for publication.
With Thanks

Reviewer 1 ·

Basic reporting

This manuscript addresses an important topic related to cadmium (Cd) stress and its impact on plant growth and physiology. The findings have potential relevance for environmental remediation strategies, particularly using okra as a phytoremediator. However, there exist several typographical, grammatical and technical errors that require major revisions.
Add key values in the abstract in terms of percentage increase or decrease.
Add 5-7 unique keywords
In introduction, the linkage between specific causes (e.g., sewage irrigation, improper fertilizer use) and cadmium contamination could be expanded with quantitative data or specific case studies to strengthen the argument.
Please provide the specific toxicological threshold levels of cadmium for plants and humans to provide a quantitative perspective.
The description of how ROS disrupts cellular homeostasis could be expanded with mechanistic insights or specific examples from prior studies.
Provide a balanced discussion of phytoremediation’s advantages and limitations. Elaborate why okra, specifically red and green cultivars, was chosen?
Previous findings on Cd accumulation in okra are briefly mentioned. Add quantitative data and compare its accumulation potential with that of other hyperaccumulators.
Clarify the study’s objectives in relation to practical applications and broader environmental challenges.
Materials and Methods: The seed sterilization method with H₂O₂ is described twice (lines 67-68 and 85-86). Consolidate the description to avoid repetition.
Provide details on how treatments were assigned to replicates and ensure randomization is mentioned explicitly.
The preparation of CdCl₂ solutions mentions dilutions (line 70) but does not specify how dilutions were prepared (e.g., intermediate steps). Include more detail on dilution procedures to improve clarity.
Clarify how data normality and homogeneity of variance were tested and justify the choice of post hoc tests.
No mention of validation or calibration for assays (e.g., ICP-MS, enzymatic assays). Include details about calibration curves, standards, or validation tests to confirm assay reliability.
In results section, incorporate statistical evidence (e.g., p-values, confidence intervals) throughout the section to validate the claims made.
Streamline repetitive phrases to improve conciseness without losing critical details.
Provide better integration of results with their potential physiological significance, keeping it brief and without overlapping with the Discussion section.
Ensure all terms are clearly defined or contextualized, particularly technical metrics like the translocation factor (TF).
Include correlation, principal component analysis or heatmap to establish a relationship between observed parameters.
The discussion repeats well-established effects of Cd on plants, such as inhibition of germination and photosynthesis, without adding significant novel insights. Focus more on unique aspects of your findings, such as the differential response between red and green okra.
The observed increase in chloroplast pigments under Cd stress in okra contradicts most literature and is not thoroughly explored. Provide more detailed hypotheses or experimental data to explain this anomaly. Was this related to stress-induced metabolic shifts, pigment stabilization, or other factors?
The variability between the two okra varieties (red and green) is mentioned but not explored in depth. Discuss possible genetic or physiological reasons for green okra's better Cd tolerance. Consider referencing comparative studies on plant varietal responses to heavy metal stress.
The conclusion emphasizes okra's potential for Cd phytoremediation without adequate justification based on the current study's results. Strengthen this claim with quantitative data, such as Cd accumulation efficiency, biomass production, or tolerance thresholds.
Specific comments:
The title of the study does not align well with its stated objective. To improve clarity and coherence, I recommend refining the title to explicitly reflect the study's emphasis on the phytoremediation capabilities of okra.
Line 1: Instructions replace with introduction.
Line 7-8: The element Cd is not essential for the growth and development of 8 plants for plant growth and development,
Line No. 20: The phrase "to Dietz lerating metal exposure" appears to be a typographical error and should be revised.
Terms like "plastic cups" (line 89) and "vessels" (line 111) are used inconsistently. Use standardized terminology to describe experimental containers.
The microwave digestion procedure lacks operational details, such as digestion temperature and duration (lines 103-104).
The phrase "obviously" (Line 198) is subjective and should be avoided in scientific writing.
Add a separate heading for conclusion section.
Correct caption of Fig. 5 "The content of chloroplastidic pigments in leaves (A) and roots (B) of red okra and green okra"
Where did the authors presented the results of translocation factor (%)?

Experimental design

Experimental design isn't seems to be robust. Why the authors only tested 50 micromolar Cd ? Lacking critical information about the maintenance of experimental conditions and randomization.

Validity of the findings

Some of the findings seems not valid, questioning the authenticity of experimentation and reporting. Figure 5 suggests chloroplast pigments are enhanced under Cd stress. H₂O₂ content increased in leaves of red okra under control conditions. The observed increase in chloroplast pigments and reduced H₂O₂ levels under Cd stress in leaves contrasts with most literature, lacking sufficient mechanistic explanation. Further investigation is needed to validate these findings.

Reviewer 2 ·

Basic reporting

the authors wrote article on: Effects of cadmium exposure on morphological and
physiological characteristics of okra (Abelmoschus esculentus L.).
This study examined the effects of cadmium (Cd) exposure on the growth and physiological responses of two okra varieties (red and green). Cd application inhibited seed germination and reduced plant biomass in a dose-dependent manner. They reported Green okra was more sensitive to Cd stress than red okra.
The article is a nice effort, however, there are a few shortcomings that must be addressed before the article can reach to publishing standard.
Overall, the quality of English in the article is weak. The excessive use of complex vocabulary and unnecessary jargon makes the language less accessible and harder to comprehend. Simplifying the wording and using clearer, more concise expressions would improve readability and ensure that the content is easily understood by a wider audience. For example:
Line 2: The intensifying problem of heavy metal pollution……… The word intensifying is unjustifiable
Line 7-8: The element Cd is not essential for the growth and development of
8 plants for plant growth and development……….the word growth and development is redundant
Line 13: oxygen species (ROS)(Rizwan……….. Space after (ROS)
Line 14: Rasafi et al. 2022)These effects ultimately………. Space
Line 15: and quality (Rizwan et al. 2017). Too much space after quality
Line 20: and to Dietz lerating metal exposure I believe you meant "Dietz rating metal" or more specifically, "Dietz rating for metal". If so please write clearly.
Line 67: Ninety full seeds of each……………….. what is meant by full seed?
Line 101: filtered through 60 mesh……"60 mesh" means the filter has 60 openings per linear inch or what? Please elaborate……………..
Line 148: in Morelli and Scarano (2004) with……………. two different fonts and sizes?
Line 261: Rao et al. 2019).For instance, …………..space after full stop.
Line 264: observed in pea (Pisum sativum)(Majeed et al. 2019) space between parenthesis
Line 267: harmala (Nedjimi 2020) . Remove space between parenthesis
Line 289: carota)(Sharma et al. 2010).Sharma et al.(2010) found ………..spaces after parenthesis

Experimental design

The study should include symptoms and effects of Cd on Okra plant. It also should include whether the areas where the red and Green Okra is being cultivated, is Cd contaminated?
I think the introduction section should include information on:
What is the average yield and production area of Red okra?
And which ethnic group consumes it?
This will explain the relative application of the study.

Validity of the findings

The experiment tries to build on already published study, however, clearly lacks to justify the scope of the study.
While the text mentions novel contributions, it's unclear what these contributions are or how they advance the field.
The text presents the current study as an improvement over Sharma et al.'s (2010) study without critically evaluating the objectives of the study. If the study was designed to merely characterize the the effect of Cd on OKra, then it should have looked into the effects of fruit or edible part of OKra that are consumed by Human beings.
If the crop is intended to be used for soil reclamation, what makes this crop better than other hyperaccumulator crops that are pereninals as well?

Reviewer 3 ·

Basic reporting

The study aims to address the impact of cadmium stress on two okra varieties, with a focus on their morphological and physiological responses. The findings contribute valuable insights into the potential of this species for phytoremediation in Cd-contaminated soils. The manuscript is written in detail, however authors need to address the following technical comments for the improvement of this manuscript before further processing.

The title, abstract & introduction show that the study is on Abelmoschus esculentus varieties (red & green okra). However, Fig 3 highlights the names of plants under study as Hibiscus coccineus and Hibiscus esculentus (seems like two species of Hibiscus, instead of varieties of a same species. This ambiguity needs clarification.
Moreover, give the complete author citation of Abelmoschus esculentus.

The abstract mentions dose-dependent inhibitory effects and a significant reduction in plant biomass however does not provide quantitative data that might enhance the clarity and scientific rigor of the findings.
The section briefly states that green okra is more sensitive to Cd stress than red okra but does not elaborate on the physiological or morphological differences responsible for this sensitivity.

Recheck the heading of first section "Instructions" and correct it accordingly.

The introduction section is too lengthy. Some concepts like the impact of Cd on plant physiology and the role of antioxidant systems are repeated multiple times. The repetition can be avoided.
Moreover, the section highlights okra as a potential Cd accumulator, however, it lacks the details on why okra is particularly suited for this role compared to other plants, especially in terms of its biomass, adaptability and phytoremediation potential.

Split the longer sentences into shorter, throughout the manuscript to improve its clarity.

Check the format of in-text citation.

Describe the abbreviations at first mentioned places, like NPSH in methodology section.

Experimental design

Add references in the methodology section to represent the authenticity of selected methods.

It is suggested to analyse the data with advanced statistical methods like two-way ANOVA or other multivariate analyses (in addition) to represent the interaction between treatments and plant varieties.

Validity of the findings

The results are written in detail. However, adding significance detail of various findings would improve this section.

In discussion section, the repetition about the findings of seed germination and plant morphology should be avoided. Consolidate the description about Cd stress effects on above parameters to improve the readability.

The observed reduction in germination rates and pigment levels due to Cd stress should be compared with the previous studies for the justification of present findings.

Highlight the unique findings of your study like the early stage chlorophyll increase and explain how these contributed to the existing knowledge. Although the potential reasons for the observed increase have been suggested, however, these could be elaborated further. Like the involvement of physiological or molecular mechanisms in this increase could be hypothesized.

The concluding paragraph should be separated from the discussion. The last paragraph of discussion represents the interesting application of okra for phytoremediation but does not relate it well with the current findings. Emphasizing how specific observed traits like, biomass, antioxidant activity etc make okra a potential crop for soil remediation would improve the concluding section.

---

## Round 0.2 · Minor Revisions

Dear Authors

The manuscript still needs a minor revision before publication. The authors are invited to revise the paper considering all the suggestions made by the reviewers. Please note that the requested changes are required for publication.

With Thanks

Reviewer 1 ·

Basic reporting

The authors have made significant efforts to enhance the quality of the manuscript. However, the manuscript still requires minor revisions to meet the standards of scientific rigor and clarity. Below are the specific recommendations for improvement:
The manuscript requires professional editing to address issues related to grammar, syntax, and typographical errors.
Include a detailed and standardized procedure for determining the moisture content of shoots and roots.
The sentence, “The experiment was set up with three replicates,” is redundant and should be removed to avoid repetition.
The discussion on chlorophyll enhancement under stress conditions requires a more in-depth scientific explanation. Instead of mere comparison, provide a mechanistic understanding of how chlorophyll levels are modulated under stress. Incorporate relevant biochemical pathways, physiological responses, and molecular mechanisms and environmental influence to strengthen the scientific basis of the findings.
Carefully review and address the anomaly in Figure 6A, where the H₂O₂ content in the leaves of control red okra appears unexpectedly elevated. Re-examine the experimental data and methodology to identify potential sources of error or biological explanations for this observation. Clarify this in the results and discussion sections.
In Lines 218–219, the procedure states: “Briefly, roots or leaves (about 500 mg each) were ground in liquid nitrogen and 1 mL of 5% (w/v) sulfosalicylic acid solution containing 6.3 mM diethylenetriaminepentaacetic acid (DTPA) was added to each sample.” To enhance reproducibility, specify the exact volume of the 6.3 mM DTPA solution used.
Ensure all methodological details are precise and unambiguous to facilitate replication of the experiments.
Line 386: Correct the typographical error in “ROS overproduction (Sch¸tzend¸bel et al., 2001; Gill & Tuteja, 2010; He et al., 2017)Cd stress” to ensure proper citation formatting and clarity.
Lines 396–397: Address spacing and punctuation errors in “adjustment and metal detoxification(Sun et al., 2007; Zayneb et al., 2015; Semida et al., 2018; Rady et al., 2019)..” Ensure consistent spacing and correct the use of full stops.

Experimental design

Good

Validity of the findings

Findings seems valid. Fig. 6A needs reconsideration.

Reviewer 2 ·

Basic reporting

The authors have now significantly improved the article. Its expression has improved form very complex or over-staement to rational write up.

Experimental design

Improved and acceptable.

Validity of the findings

Now justified their intent and is acceptable now.

Reviewer 3 ·

Basic reporting

The revised version of manuscript has been carefully reviewed. Authors have well addressed all the concerns raised in the previous review. This major revision has significantly enhanced the clarity, rigor and overall quality of the study.
The manuscript is now well-written with clear and professional language. The introduction provides sufficient background representing the problem statement and novelty of the study. The objectives of the study have been clearly defined.

Experimental design

The experimental procedures have been well-documented, ensuring reproducibility. Appropriate references have been added representing the authenticity of the methods. The authors have provided adequate details in this section, including the selection of okra varieties, experimental setup and stress conditions.

Validity of the findings

The authors have successfully addressed previous concerns regarding data interpretation, enhancing the credibility of their findings. The comparative approach between red and green okra varieties under cadmium stress has provided valuable insights into plant stress physiology and phytoremediation.

Additional comments

The revised manuscript presents a well-supported scientific contribution. It could be a significant addition to this journal.

---

## Round 0.3 · accepted · Accept

Dear Authors,

I am pleased to inform you that the manuscript has improved after the last revision and can be accepted for publication.

Congratulations on accepting your manuscript, and thank you for your interest in submitting your work to PeerJ.

With Thanks

Reviewer 1 ·

Basic reporting

I am pleased to see that the authors have satisfactorily addressed most of the queries raised in the previous round of revision. The manuscript is now suitable for publication.

Experimental design

Experimental design is promising

Validity of the findings

Findings seems valid.